# Dynamical order and many-body correlations in zebrafish show that three is a crowd

Alexandra Zampetaki [1,2] ✉, Yushi Yang [3] ✉, Hartmut Löwen[2] & C. Patrick Royall [4] ✉

Zebrafish constitute a convenient laboratory–based biological system for studying collective behavior. It is possible to interpret a group of zebrafish as a system of interacting agents and to apply methods developed for the analysis of systems of active and even passive particles. Here, we consider the effect of group size. We focus on two– and many–body spatial correlations and dynamical order parameters to investigate the multistate behavior. For geometric reasons, the smallest group of fish which can exhibit this multistate behavior consisting of schooling, milling and swarming is three. We find that states exhibited by groups of three fish are similar to those of much larger groups, indicating that there is nothing more than a gradual change in weighting between the different states as the system size changes. Remarkably, when we consider small groups of fish sampled from a larger group, we find very little difference in the occupancy of the state with respect to isolated groups, nor is there much change in the spatial correlations between the fish. This indicates that fish interact predominantly with their nearest neighbors, perceiving the rest of the group as a fluctuating background. Therefore, the behavior of a crowd of fish is already apparent in groups of three fish.

Collective behavior is fundamental to systems as diverse as group living in animals[1,2], flocking in artificial active matter such as colloids[3] and the well-known phenomena of phase transitions in condensed matter[4]. We know from statistical mechanics of physical systems that interactions between constituents should underlie such collective behavior, which may be applied in a biological context to phenomena such as condensation in midges[5,6], schooling, milling and swarming in fish[7–9], coordinated turning in flocks of birds[10,11], clustering in penguin huddles[12,13] and the collective movement of sheep[14–16]. In fact, the ultimate goal of many recent studies on the collective behavior in animal systems is to infer the interactions from available experimental trajectories, and construct reliable models for their behavior[7,17–21].

In physical systems, the relationship between constituent interactions and their spatial correlations is precisely defined[22] so that

characterization of these interactions is possible through measurements of two-body and higher-order correlations. Although higher–order correlations can be important at high densities where individuals are strongly coupled, at low density Kirkwood superposition which presumes that 3-body correlations are captured by 2-body correlations is highly accurate[22]. In the context of animal behavior, social forces, as a proxy to the many–body correlations, have been inferred in previous studies[7,17,18,23–25]. In the case of fish, this may be done either using the force map technique[7,17,20] or sophisticated fitting techniques of relevant observables[19,21]. Often, such a force inference is done for two-fish systems, thought to be the minimal system allowing us to gain an understanding of the complexities of many–fish behavior. However, we still do not fully understand the effect of many–body correlations in animal groups, for the lack of

¹Institute for Applied Physics, TU Wien, A-1040 Wien, Austria. ²Institut für Theoretische Physik: Weiche Materie, Heinrich-Heine-Universität, 40225 Düsseldorf, Germany. ³HH Wills Physics Laboratory, Tyndall Avenue, Bristol BS8 1TL, UK. ⁴Gulliver, UMR CNRS 7083, ESPCI Paris, Université PSL, 75005 Paris, France. ✉e-mail: zampetaki@iap.tuwien.ac.at; yangyushi1992@icloud.com; paddy.royall@espci.fr

direct determination even though the relevant methods to measure such correlations are well established[26–28].

Another phenomenon, which is exhibited by systems small in comparison to the thermodynamic limit of a very large number of interacting agents is intermittency between multiple states. In equilibrium, in the thermodynamic limit, outside phase coexistence, a system will adopt a single state. In contrast, small systems can exhibit intermittency between multiple states[29,30]. Intermittent behavior can be found also in large out of equilibrium systems, for example in the case of active matter[31–33]. The number of constituents in biological systems is necessarily small in a thermodynamic sense and by analogy the role of system size and fluctuations can have significant consequences[34], such as variations in the speed of different fish[16,35–38]. Indeed, a bistable state was observed for zebrafish, where the schooling and shoaling states coexist[8,39]. Furthermore, a system of golden shiner fish with a tristable state has been observed, as the fish group exhibits a milling state in which individuals rotate around the group center[9]. This intermittent, multi-stable feature is also observed in other animal species[40–42], and it has been reproduced in computer simulations of different agent-based models[43–45]. However, how these interactions scale with group size is poorly understood, likewise the relationship of group size and collective behavior is not extensively explored[9].

Here, we use methods from statistical mechanics of many-body systems to systematically investigate higher-order correlations and multistable states in a model system of zebrafish. We consider system sizes of $N = 2, 3, 4,$ and 50 fish, by reconstructing their 3D trajectories with a custom 3D tracking system[46]. Such an approach allows us to probe changes in the correlations between the fish across the group sizes. That the correlations we observe are quite well reproduced with a pairwise model shows that triplet and higher-order interactions are small, justifying the use of such a model. For a group of $N \geq 3$ zebrafish, we observe the existence of schooling, milling and swarming states, which have been seen in other fish species[8,9,41]. We find that groups of three fish exhibit transitions between these states and thus may be said to exist in a tristable state. This behavior can be reproduced with a simple agent-based model. We examine larger groups up to $N = 50$, and find a gradual emergence of the dominance of the swarming state, but there is no other effect of group size under our analysis. Locally, however, sub-systems of 3 fish within larger systems exhibit behavior which is hard to distinguish from that of isolated groups of 3 fish. This last point suggests that the fish interact predominantly with their nearest neighbors with more distant fish acting as a background.

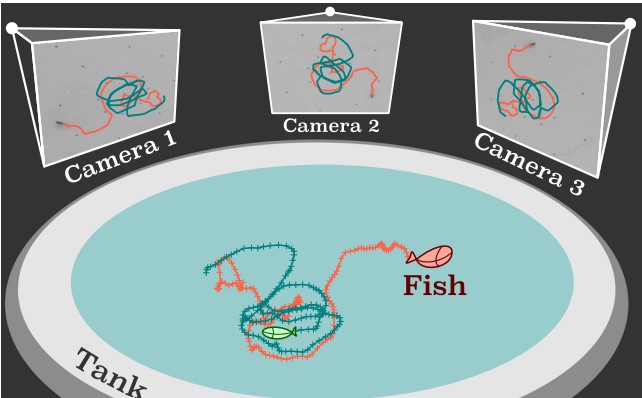

**Fig. 1 | Experimental set-up.** 3D reconstruction of fish trajectories with 3 synchronized cameras. The locations of the fish were determined in each 2D image, to calculate the 3D coordinates. The 3D coordinates are subsequently linked into 3D trajectories.

## Results

Groups of between 2 and 50 wild type zebrafish were observed in a bowl-shaped tank with parabolic section. As indicated in Fig. 1, the geometry was selected because we want to measure the 3D behavior of the zebrafish, since the fish naturally swim in 3D in rivers[47]. The movement of the zebrafish was recorded with three synchronized cameras, illustrated in Fig. 1. In the images from different cameras, we calculate the 2D feature points belonging to the different fish. Knowing the intrinsic parameters and the positions of the cameras from a calibration process, we reconstructed the 3D coordinates of the fish and linked these coordinates into trajectories. We collected datasets from the same group of adult zebrafish repeatedly, to minimize the variance in the behavior among different groups[48]. Further details are given in the Methods and ref. 46. We also implement a minimal agent-based model in which the interactions between the fish are parameterized, as we discuss below.

### Order parameters

We characterize the dynamical states of the fish with two order parameters. The polarization order parameter $O_p$, is defined as

$$O_p = \frac{1}{N} \left| \sum_{i=1}^{N} \frac{\mathbf{v}_i}{|\mathbf{v}_i|} \right|, \tag{1}$$

where $\mathbf{v}_i$ is the velocity of the $i$th fish. The value of $O_p$ will be $\approx 1$ if all the fish swim in the same direction, and will be $\approx 0$ if the swimming directions of the fish are random. Similarly to ref. 9, we call the states with a high $O_p$ value (here $O_p > 0.65$) schooling. We also define the rotation order parameter $O_r$, as

$$O_r = \frac{1}{N} \left| \sum_{i=1}^{N} \frac{\mathbf{v}_i}{|\mathbf{v}_i|} \times \left( \frac{\mathbf{r}_i - \mathbf{r}_{cm}}{|\mathbf{r}_i - \mathbf{r}_{cm}|} \right) \right| \tag{2}$$

where $\mathbf{r}_i$ is the location of the $i$th fish, and $\mathbf{r}_{cm}$ is the location of the group center. For a group of fish rotating around the same axis, the value of $O_r$ will be $\approx 1$. We term the states with a high $O_r$ value (here $O_r > 0.65$) milling. Disordered states with low $O_r$ and $O_p$ values (here $O_p, O_r < 0.35$) are termed swarming. Note that these order parameters require the instantaneous velocities of the fish, obtained by tracking the fish between two frames, rather than extended trajectories of each fish. Moreover the spatial correlations that we present require only instantaneous snapshots. Throughout, we set the unit of length to be the average body length of the fish, 30 mm.

In most of our analysis, we neglect differences between individuals. However, our agent-based model allows us to probe the effect of a velocity distribution in the fish population as opposed to a constant velocity for each agent. In some situations, this can be significant[16,35–38]. We have investigated the effects of such a distribution to which we return below.

### Experimental results

Our experimental set-up is indicated schematically in Fig. 1, from which we obtain different trajectories of up to 50 zebrafish. We measure spatial correlations between the fish as shown in Fig. 2. Here, Fig. 2a defines the orientation, range and velocities of two interacting fish in the laboratory frame. In Fig. 2b, the same two fish are considered, this time in the comoving reference frame of fish $i$. Henceforth, we consider this latter representation and turn to consider 2-body spatial correlations. We shall consider groups of different numbers of fish and these we term $\mathcal{F}_N$ where $N$ is the number of fish. When we consider the local sub-group of a fish and its $n-1$ nearest neighbors in a larger group $N$, we shall write $\mathcal{F}_N^n$. Beginning with two fish, $\mathcal{F}_2$, we see that there are three peaks, corresponding to typical configurations where one fish is following another (the front peak which we have labeled $\beta$), and the configuration where the two fish are swimming side-by-side (the two side peaks which we have labeled $\alpha$ and $\gamma$). When we consider

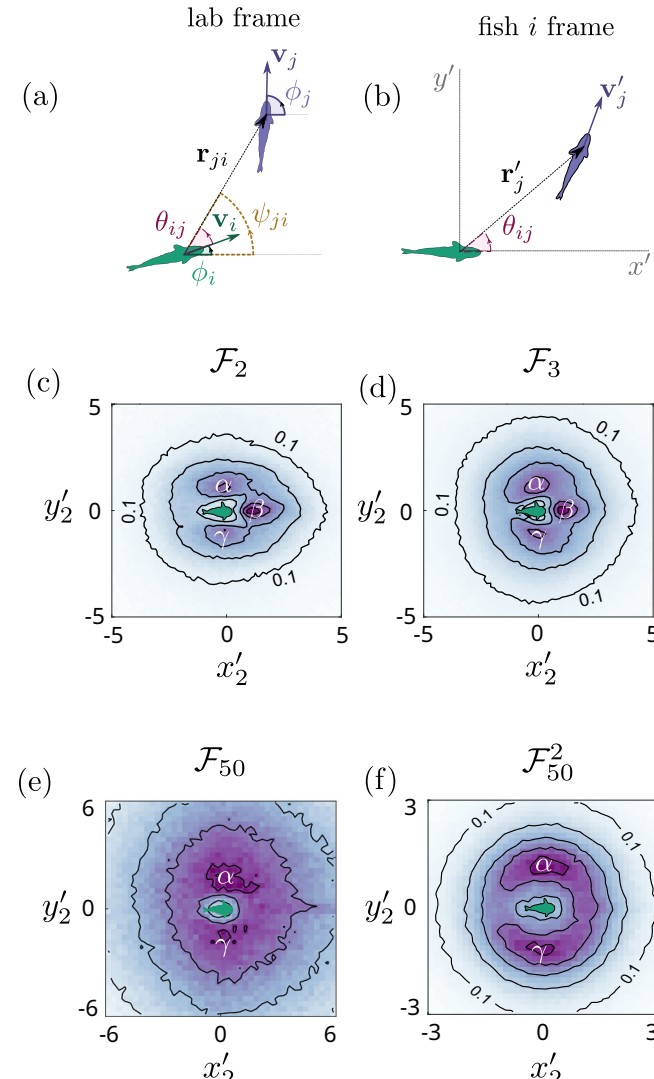

**Fig. 2 | Pair correlations in the zebrafish system.** Sketch of two fish, showing the relevant quantities for this study in the laboratory reference frame (**a**) and in the reference frame of fish $i$ (**b**). By laboratory reference frame, we mean measuring the positions and velocities of the fish with respect to a fixed point in the lab. The fish reference frame referred to in (**b**) corresponds to measuring positions and velocities with respect to the fish indicated at the origin. Average pair correlations $g_2$ in the reference frame of one of the fish for (**c**) $N = 2$ fish ($\mathcal{F}_2$), (**d**) $N = 3$ fish ($\mathcal{F}_3$) and (**e**, **f**) $N = 50$ ($\mathcal{F}_{50}$) fish. In (**e**) the correlation of an arbitrary pair out of the 50 fish is shown, whereas (**f**) shows the correlation of the nearest neighbor pairs ($\mathcal{F}_{50}^2$). The letters $\alpha, \beta, \gamma$ mark the positions of the different peaks: $\alpha, \gamma$ indicate the side peaks in the $g_2$ and $\beta$ the front peak. Here for clarity, we have normalized the correlation distributions by the maximum value, so that the colorbar goes from 0 (white) to 1 (maximum saturation). The contour lines correspond to 0.1, 0.3, 0.5, 0.7 and 0.9. All the data for this figure are taken from the experiments.

three fish, $\mathcal{F}_3$, a similar pair correlation emerges, again with three peaks [Fig. 2d]. In the case of 50 fish, we use two representations. Figure 2e shows the same $g_2(x', y')$ representation, while in Fig. 2f, we show the spatial distribution of the nearest fish to fish $i$ at any moment. In the $\mathcal{F}_{50}$ system, the $\beta$ peak is lost, suggesting that the fish tend not to follow one another. The emergence of the different peaks, corresponding to qualitatively different swimming configurations, indicates that the fish pass through different states, as we will explore in more detail below.

**The behavior of three fish is like that of many fish**

We now consider a system of three zebrafish, $\mathcal{F}_3$ in Fig. 3a–d. For geometric reasons, this is the minimum number of fish required to

exhibit the three dynamical states of schooling, milling and swarming. These are shown schematically in the first row, and example experimental trajectories corresponding to each state are shown in the second row. If we consider the time-evolution of the order parameters $O_p$ and $O_r$ [Fig. 3d] we see that the three fish system is effectively a tristable state, by which we mean that the system transitions between each of the three states schooling, milling and swarming. This three state behavior is shown in more detail in Supplementary Movies 1–4.

We now consider the effect of system size on the multistate behavior in Fig. 3e–h for the $\mathcal{F}_{2,3,4}$ and $\mathcal{F}_{50}$ systems. The time–evolution of these systems is shown in the Supplementary Movies. For the two–fish system $\mathcal{F}_2$ (Supplementary Movie 1), indeed the swarming state (low $O_p$, low $O_r$) is not seen. However for three fish $\mathcal{F}_3$, it is found, although rather rarely (Fig. 3f, Supplementary Movie 2). As the number of fish increases, the weight of the swarming state increases at the expense of both the schooling and milling states so that for 50 fish $\mathcal{F}_{50}$, it dominates (Fig. 3g, h, Supplementary Movies 3, 4). This observation is consistent with previous results[8,9]. This increase in the swarming state appears to be gradual with no indication of anything like a "phase transition". See Supplementary Fig. 2 where we plot intermediate system sizes for simulated data. In other words, at the level of the order parameters $O_p$ and $O_r$, the characteristics of the "crowd" (here the 50 fish system) are already present in the 3 fish system, but not in the 2 fish case.

**Local subgroups in a large system behave like isolated groups**

We can also analyze local subgroups of larger systems. That is to say, we determine the order parameters ($O_p$, $O_r$) for the subgroups within the $\mathcal{F}_{50}$ system, which we define as the fish of interest and its nearest neighbors. In Fig. 3i–k, we show the $\mathcal{F}_{50}^{2,3,4}$ systems. The similarity with the isolated groups $\mathcal{F}_{2,3,4}$ is quite remarkable. While the absolute values of the populations in each state vary slightly, the trend is identical. In other words, at the level of this analysis, the behavior of $n = 2, 3, 4$ fish is essentially unaffected by more distant neighbors, which might suggest that the fish interact weakly over longer ranges, which is consistent with earlier work on mosquitofish which showed that the nearest neighbor dominates the interaction[7]. The results from these dynamical order parameters then contrast somewhat with the spatial order parameters shown in Fig. 2, where we see that the $\beta$ lobe of the $\mathcal{F}_{50}$ system is weaker than that in the case of $\mathcal{F}_2$ (Fig. 2c).

To probe this intriguing behavior further, we consider spatial correlations between the fish in more detail. In Fig. 4a1, a2, a3, we show $g_2(x_2', y_2')$ for the different states of the 3 fish system. Here, we see that in the schooling state, the $\beta$ peak is quite strong, as the fish follow each other, while in the milling state the $\alpha$ and $\gamma$ peaks dominate. This is reasonable, since in the milling state the three fish rotate around their center of mass, so they are more likely to be found at the sides of a reference fish, with an opposite orientation from it. For the swarming state, the $\alpha$ and $\gamma$ peaks dominate, but there is a weak $\beta$ peak due to the randomness of the motion. These results are consistent with the $g_2(x_2', y_2')$ behavior with respect to system size shown in Fig. 2, where we saw that small systems had significant $\alpha, \beta$ and $\gamma$ peaks, indicating multistability between schooling, milling and swarming.

For the local subgroups of larger system sizes, here $\mathcal{F}_{50}^3$, the pair correlations $g_2(x_2', y_2')$ for the milling and swarming states (Fig. 4b2, b3) are similar to those for the 3 fish system (Fig. 4a2, a3), showing an increased weight of the side peaks $\alpha, \gamma$. The schooling state, however, is quite different from the $\mathcal{F}_3$ system, showing a domination of the $\alpha, \gamma$ peaks over the front peak $\beta$. Note that in this case the local subgroup of 3 nearest neighbor fish is highly polarized and the side peaks point to the fish swimming side-by-side with the same orientation, unlike the side peaks for the milling state. This implies that, during schooling, the local subgroups of larger groups swim parallel to each other instead of following each other in a line, forming more compact dynamical configurations, probably due to the limitation of available space.

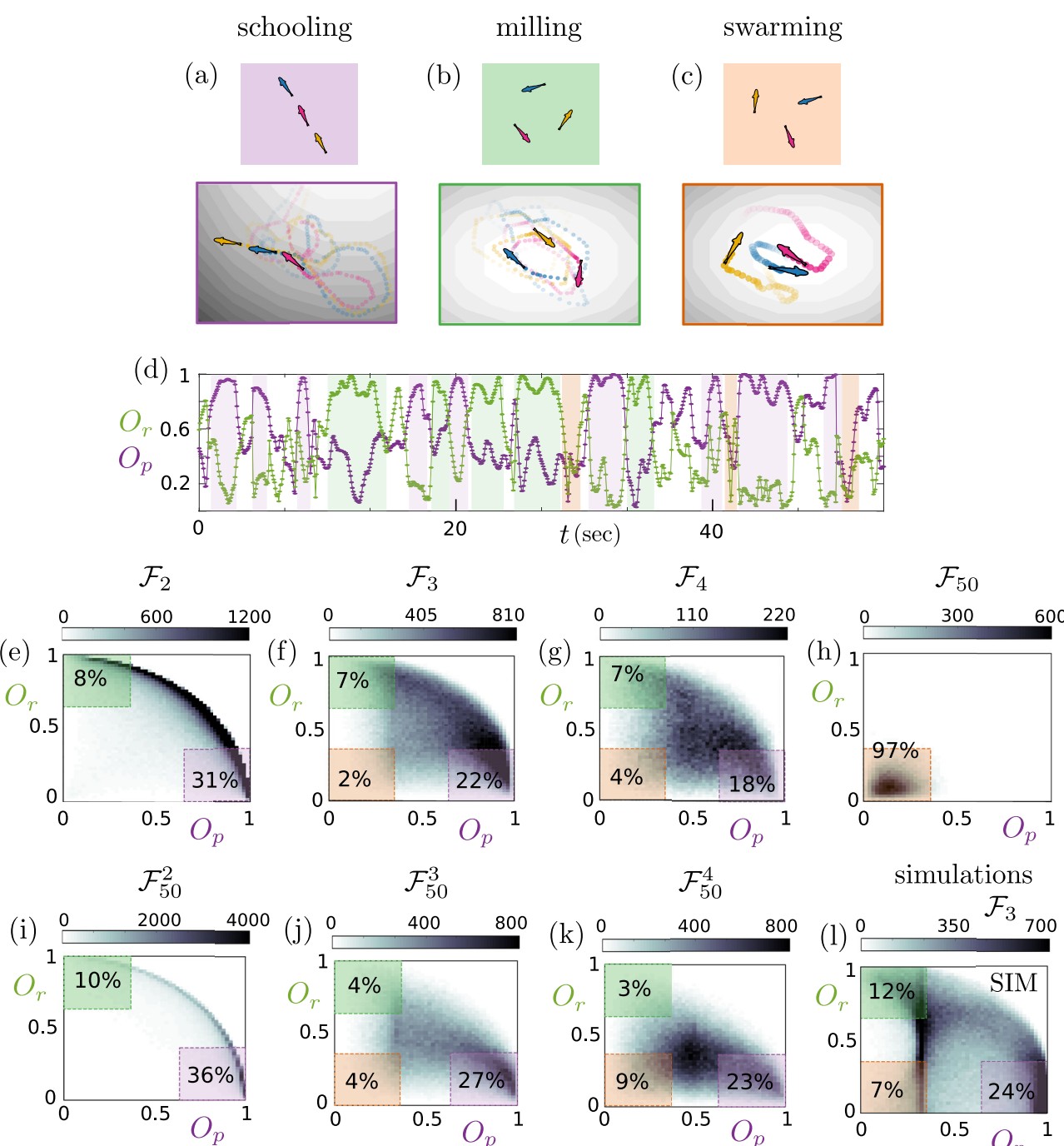

**Fig. 3 | Three fish exhibit schooling, milling and swarming states.** Sketched and example experimental trajectories of the different dynamical states of three fish (**a**) schooling, (**b**) milling and (**c**) swarming. **d** Time evolution of the rotational and polarization order parameters $O_r$ and $O_p$ respectively. Here, we consider a 3-fish trajectory. The times at which the different states are found are marked with the corresponding color. Density plots of $O_r$ versus $O_p$ for (**e**) $N = 2$ fish ($\mathcal{F}_2$), (**f**) $N = 3$ fish ($\mathcal{F}_3$), (**g**) $N = 4$ fish ($\mathcal{F}_4$) and (**h**) $N = 50$ fish ($\mathcal{F}_{50}$), indicating the occurrence of the different states. Density plots of $O_r$ versus $O_p$ for small local groups of $N = 50$ fish, consisting of $n$ nearest neighboring fish ($\mathcal{F}_{50}^n$): (**i**) $n = 2$ ($\mathcal{F}_{50}^2$), (**j**) $n = 3$ ($\mathcal{F}_{50}^3$) and (**k**) $n = 4$ ($\mathcal{F}_{50}^4$). **l** Density plot of $O_r$ versus $O_p$ for $N = 3$ fish from simulation data. The colored rectangles mark the regions of $O_p$, $O_r$ values which indicate the occurrence of each state, with the enclosed values showing the percentage of cases that exhibit the respective state. Here the gray shading refers to the count in each bin, as expressed by the colourbar above each panel. The bins have a side of 0.025.

More information about the many-fish configurations can be gained by considering higher-order spatial correlations. Here we examine first the probability distribution of the bonding angle between the three fish $\theta_r^{(3)}$ shown in Fig. 4c. Here, we set the range of separation between the fish to the principle peak of $g_2$ for each state (See the SI for further details). As we see, for the $\mathcal{F}_3$ system the schooling state shows a peak at smaller angles ($\theta_r^{(3)} \approx 30°$) and an increasing probability for high angles around 180°, pointing once

again to the fish following each other. In contrast, for the milling state there is a single peak at ($\theta_r^{(3)} \approx 60°$), as expected by symmetry, whereas the result for the swarming state lies between the milling and schooling ones. Such a discrepancy between the different states is lost when exploring the local subgroup of a larger system $\mathcal{F}_{50}^3$, indicating a suppression of spatial correlations.

Among the key achievements using higher-order spatial correlations is Kirkwood superposition, where three-body correlations are

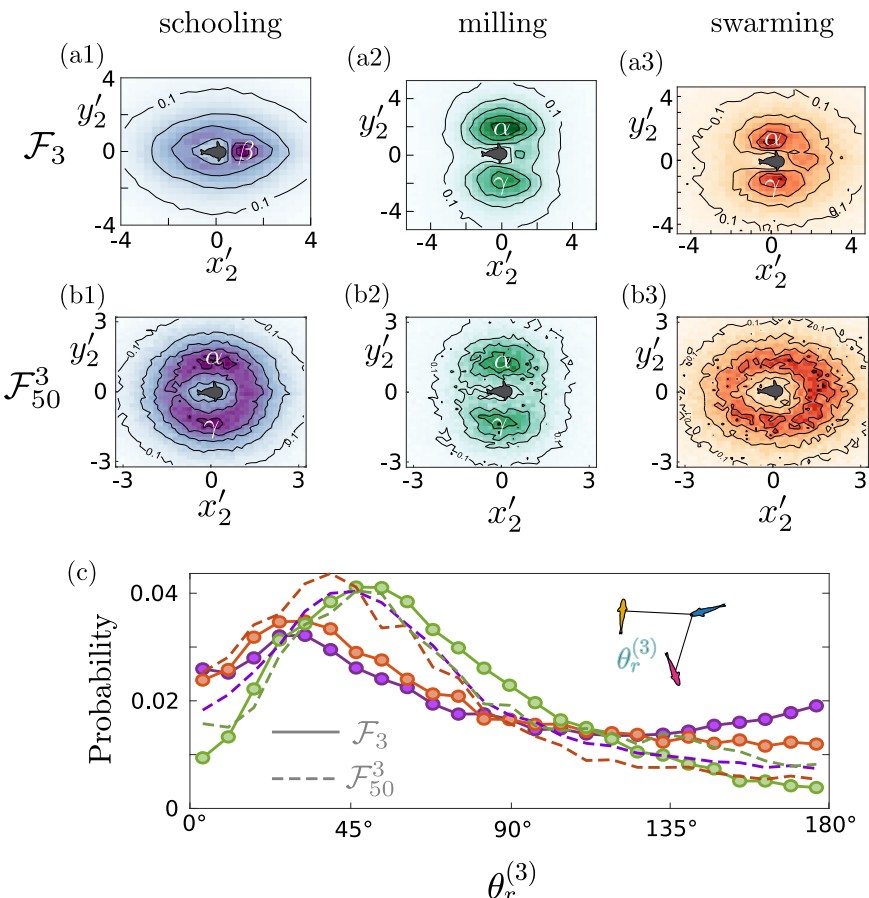

**Fig. 4 | Pair correlations $g_2$ in the reference frame of one of the fish for the different states. a1–a3** $N = 3$ fish ($\mathcal{F}_3$) and (**b1–b3**) local groups of $n = 3$ nearest neighbor fish in the case of $N = 50$ fish ($\mathcal{F}_{50}^3$). We distinguish between the different states as follows (**a1**, **b1**) schooling, (**a2**, **b2**) milling and (**a3**, **b3**) swarming state. Here for clarity, we have normalized the correlation distributions by their maximum value, so that the colourbar goes from 0 (white) to 1 (maximum saturation). The contour lines correspond to 0.1, 0.3, 0.5, 0.7 and 0.9. **c** Probability distribution of the three-fish bond angle $\theta_r^{(3)}$ of $N = 3$ fish in the milling (green line with circles), the swarming (orange line with circles) and the schooling (purple line with circles) states. The dashed lines show the corresponding results for $\mathcal{F}_{50}^3$.

predicted from two-body correlations. Specifically, $g_3(r_{12}, r_{23}, r_{31}) \approx g_2(r_{12})g_2(r_{23})g_2(r_{31})$, with $r_{ij}$, the distance between the fish $i$ and $j$. This superposition principle was originally suggested by Kirkwood[49] and has since become a cornerstone to approximate triplet correlations in liquid state physics[50]. Kirkwood superposition holds in dilute, weakly interacting passive systems and is expected to hold even in non-equilibrium steady states as in the present case if three-body interactions are weak. To our knowledge, it has not been probed in active matter or biological systems exhibiting collective behavior until now, and there is no reason to suppose that it would hold. In Fig. 5, we show that for the three fish system, there is a discrepancy between Kirkwood superposition and the measured $g_3(\tilde{x}_3, \tilde{y}_3)$ in the schooling state. In particular, there is a high probability of a third fish between two fish in the actual data compared to the prediction from Kirkwood superposition, as shown in the difference between Fig. 5b1, a1, which could imply the existence of some three-body interactions in the case of three fish in the schooling state.

Here for a better visualization of the three body correlation, without loss of generality we constrain the distance of one pair, $r_{12}$, within a small window around its most probable value $r_{max}$. Then, assuming the $x$-axis to coincide with the orientation of $\mathbf{r}_{12}$, and the origin with the midpoint $\mathbf{r}_O = (\mathbf{r}_1 + \mathbf{r}_2)/2$, we find the 2D distribution of the third fish position $g_3(\tilde{x}_3, \tilde{y}_3)$. A similar procedure is followed for the visualization of the Kirkwood superposition $g_3^{(K)}(\tilde{x}_3, \tilde{y}_3)$. More details can be found in the SI and in ref. 51.

In Supplementary Fig. 4, we show that Kirkwood superposition holds well in the milling and swarming states suggesting that in these states, three-body interactions are weak. To probe further the discrepancy in the schooling state, we consider different configurations in Fig. 5. We find that in fact when two of the fish swim side–by–side parallel to each other, Kirkwood superposition holds rather well (Fig. 5a2, b2), predicting that the third fish either leads or follows the two fish. The discrepancy of the third fish between the two emerges when the fish follow one another in a line as shown in Fig. 5a3, b3. This is not the case for the subgroups $\mathcal{F}_{50}^3$, where even when the two fish follow each other the third fish swims parallel to them on the side (Fig. 5c3).

## A minimal model

To capture the behavior of the zebrafish, we simulate an agent–based perception model. For simplicity, at first we give each agent a constant speed $v_0$, before considering the case of variable speed. Our model is implemented in 2D which takes the confinement of the tank in the lateral plane (a circle) into account and includes strong attractions and weak aligning interactions between the fish, supplemented with a repulsion at short distances. Furthermore, we implement a field of view between the fish. As previously found, for zebrafish the attraction plays a significant role in their behavioral interactions[19,21]. We have observed that in simulations dominated by such attractive interactions, it is challenging to detect a prominent milling state as found here (Fig. 1). In order to overcome this difficulty we have also taken into account in our model a form of hydrodynamic interactions, in the spirit of ref. 52. Further details of our model which exhibited all three states are provided in the Methods and SI. Unless explicitly stated, the

schooling

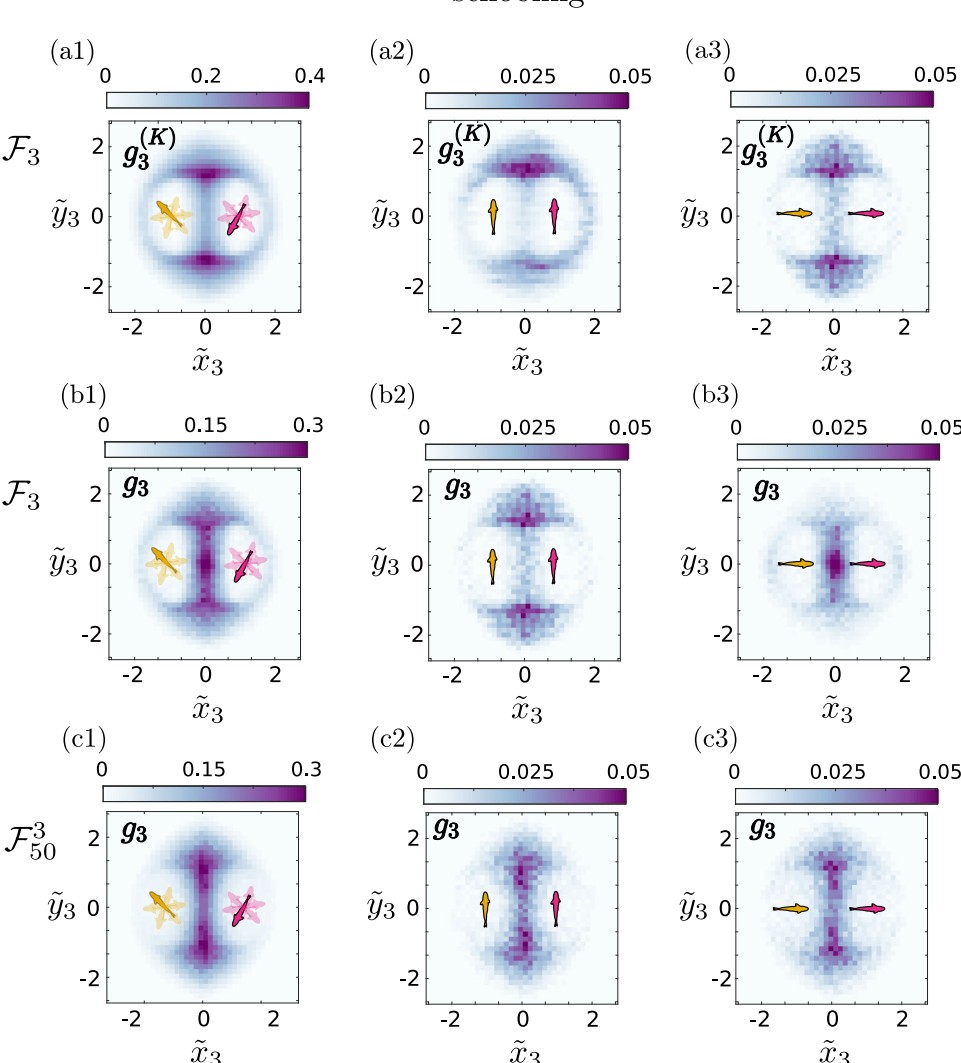

**Fig. 5 | Analysis of three-body correlations in zebrafish. a1–b3** show data from $N = 3$ fish ($\mathcal{F}_3$) and (**c1–c3**) local group of $n = 3$ fish ($\mathcal{F}_{50}^3$) fish in the schooling state as found in the experiment. The first row (**a1–a3**) shows the Kirkwood superposition approximation of the three fish correlation $g_3^{(K)}$ for $\mathcal{F}_3$. The second row (**b1, b2, b3**) shows the full three fish correlation $g_3$ for $\mathcal{F}_3$ and the last row (**c1, c2, c3**) shows $g_3$ for $\mathcal{F}_{50}^3$. In (**a1, b1, c1**) we present spatial three-body correlations $g_3^{(K)}, g_3$, without resolving the orientation of the two reference fish. In (**a2, b2, c2**) we present the three-body correlations $g_3^{(K)}, g_3$ for the reference fish aligned side-by-side whereas in (**a3, b3, c3**) we show $g_3^{(K)}, g_3$ for the reference fish aligned head-to-tail.

simulations were carried out with the set of parameters in Supplementary Tables I and II for the fixed and variable speed models respectively.

As shown in the SI, we find that our model predicts correctly the intermittent dynamics observed already for three fish, exhibiting schooling, milling and swarming (Supplementary Fig. 1). It shows also that by increasing the number of fish, the system spends progressively more time in the swarming state and that local subgroups of larger system behave like isolated groups. Regarding the pair correlations, our simulation results capture qualitatively well the fact that there are prominent front-and-back peaks in the schooling state, whereas the highest peaks in the milling state are those at the sides of the reference fish (Supplementary Fig. 3). The simulations also show an increased probability to find the third fish between the two reference fish, when the latter swim in a head-to-tail configuration (Supplementary Fig. 5). However, in the simulations this is mostly captured by the Kirkwood superposition, and there is not such a prominent middle peak as in the experiments (Fig. 5b3). This fact could imply the existence of some three-body interactions in the three-fish shoals in the experiments. Overall our simulations show somewhat weaker differences between

the local subgroups of larger groups $\mathcal{F}_N^n$ and the corresponding isolated fish groups $\mathcal{F}_n$ than that found in experiments.

So far, we have assumed implicitly in our analysis of the experiments and explicitly in our simulations that all the fish are identical. Of course, in biological systems (unlike many, but not all[53] physical systems), this is not the case. Variable speed impacts collective movement[36,38], leadership behavior is encountered[36,37], along with differences in some fish which are less risk-averse than others[35]. Here, we probe the effect of a variation in the distribution of speeds of individuals due to the social interactions and boundary (see SI). In fact, we find this has rather little effect on the observables we study, as shown in Supplementary Fig. 6.

An important reason for the observed discrepancies between experiment and simulation could be the simplified assumption of tank shape as a 2D circle instead of a parabolic tank (see also discussion in SI). Further limitations come from the assumption of an overdamped dynamics and the simplified description of the interactions. It would be interesting therefore to explore in further numerical studies whether more sophisticated models can improve the agreement with the experimental results presented here.

## Discussion

We have considered groups of 2–50 zebrafish and characterized their behavior in terms of dynamical order parameters and spatial two- and three-body correlation functions. We further investigate the behavior of smaller local sub-groups of a larger 50 fish group. The minimum group that can exhibit the three states characteristic of the zebrafish system is three fish, and in fact, we find that this small system features most of the phenomena of larger groups, at the level of our analysis.

The main effect of increasing system size is to shift the statistical weight between the three states of the multistate system that the fish form. In particular, for small systems, the dominant states are the more ordered schooling and milling states, with the disordered swarming state exhibiting a smaller weight. Upon increasing the system size, the fish spend more and more of their time in the swarming state, with no evidence for any kind of sharp transition. Thus, remarkably, we find that, under our analysis, the basic characteristics of large groups are captured by as few as three fish.

This system size dependence would appear to be an analogue of the approach to the thermodynamic limit, well known from passive systems in equilibrium. Examples of this behavior include small atomic clusters which can exhibit structure quite different from the thermodynamic limit of a crystal (for example, five-fold symmetry)[54] and confined colloidal systems which can exhibit bistable states with very different structure from the thermodynamic limit[29]. Here, however, the behavior of the large system is largely recovered with just three constituents.

We show that smaller subgroups of larger groups are very similar indeed to isolated systems of the same size. In this sense, the more distant fish could be thought of as a background, i.e., a kind of "fish soup". That is to say, the similarity of near–neighbor correlations in the small groups embedded in larger groups and the isolated groups suggests that the nearest neighbor interactions dominate and the other fish are less important. This behavior in zebrafish is different from that of starlings, which exhibited scale–free correlation[55], so that the behavioral change of an individual bird can be quickly propagated to the entire flock[10]. Instead, the behavior of zebrafish seems similar to that of the midges, because both systems turn out to exhibit disordered collective motion[5] which can be described by a global force–field neglecting the explicit interaction between individuals[56,57].

In our work, we also emphasize the importance of properly considering the many-body correlations in animal systems. As shown, the sampling of different trajectories of even small-sized groups of fish can give rise to mixed correlations and observables, due to their intermittent dynamics. In turn, each of the occurring dynamical states (schooling, milling, swarming) is associated with significantly different correlations and observable values. Our work opens the use of many–body spatial correlation functions for active matter, which might be applied to well-studied systems, for example, midges[6], and more complex systems such as birds[10,55] which may interact with more neighbors and in fish under different conditions which may cause changes in underlying social interactions[35]. An important test of these ideas would be to apply a similar analysis to well–controlled, and perhaps tunable experimental systems, such as active colloids[3].

In a biological context, our work suggests that the fish interact mainly with their nearest neighbors. Therefore, one would expect from such an analysis that the fish system would respond locally to a change in the environment, such as the appearance of a predator[35,58]. However, the observations we have made pertain to the case where the environment remained unchanged, and we have no way of knowing if the social interactions between fish would be the same. Clearly, more work is needed to address this question, perhaps by employing an approach similar to that used in ref. 24, where the shape of fish shoals under predation was explored. Given the change in shape of the shoals found in that work, there is cause to imagine that predation could change the interactions in the zebrafish system considered here, suggesting that application of the analysis we have carried out here would yield interesting insights into the social interactions of the fish. Other intriguing possibilities include relating the properties of different species to the observables considered here. For example, golden shiners have a somewhat different visual field to the zebrafish considered here[59]. The analysis we introduce here provides a new way to relating changes in social interactions to the physical properties of the fish.

## Methods

### Fish husbandry

All the fish used in the experiment were wildtype zebrafish, and they were bred at the fish facility of the University of Bristol. The fish were kept at the standard living conditions[60]. The experiments were approved by the local ethics committee (University of Bristol Animal Welfare and Ethical Review Body, AWERB) and given a UIN (university ethical approval identifier, UB/19/050).

### Experiment

The zebrafish were transferred from their living tank to a temporary tank, then introduced in the observation tank. We perform two "two-fish" experiments, and one "three-fish" experiment successively. Typically, three fish (fish A, B, and C) will be transferred to a temporary tank. The fish A and B will be introduced to the observation tank, where we carried out the first two-fish observation. Then fish B is taken back to the temporary tank, while fish C is introduced to the observation tank, so that the second two-fish experiment could be carried out. Finally, all the fish were placed in the observation tank, and we perform the three-fish observation. For the 4 and 50 fish experiment, we placed 4 or 50 fish into the observation tank from the temporary tank directly. During the observation, we discard the first 10 min to allow the fish to become familiar with the new condition, and start video recording after this period.

The experimental data were sampled as follows. For the two fish experiment, we selected 12 different pairs randomly chosen from a group of 50, and observed each pair for 1 h. For the three fish experiment, we selected 6 different triplets from a group of 50, and observed each triplet for 1 h. For four fish experiment, we selected 5 different quadruplets from a group of 50, and observed each quadruplet for 1 h. The group of 50 itself was observed for 1 h. Further details may be found in the SI.

### Data processing

The movement of the zebrafish were filmed in a separate bowl-shaped observation tank whose radius $r$ increases with the height $z$ following $z = 0.73r^2$. The water in the observation tank is the same to the water that the fish live in, but the observation apparatus has its own water circulation and filtration system. The temperature of the observation tank was heated by two commercial heaters, having a temperate of around 25℃. Three cameras (Basler acA2040 um) were used to record the movement of the fish, which were triggered by synchronized signals from an Arduino microchip[46].

From the videos, we locate individual fish in successive frames, as 2D images. With the 2D locations from different views, we calculate their 3D locations following conventional computer vision method[61,62], with the water refraction being explicitly considered[46]. The locations were linked into trajectories following a four frame linking procedure[63]. These trajectories were further linked following Xu[64].

### Simulation model

To probe in more detail the surprising behavior that we see (weak interactions inferred from dynamical order parameters, strong interactions inferred from spatial order parameters), we now turn to a perception model in which the zebrafish interactions are dominated by the repulsion-attraction between the fish. For completeness we also

include weak alignment interactions, as well as some hydrodynamic interactions, inspired by ref. 52. Our model is as follows. We assume that the fish move in 2D with a constant speed $v_0$ and can only change their velocity orientation given by the angle $\phi_i$ (Fig. 2a). Thus, the 2D overdamped equations of motion for fish $i$ read

$$\dot{x}_i = v_0 \cos \phi_i \tag{3}$$

$$\dot{y}_i = v_0 \sin \phi_i \tag{4}$$

$$\dot{\phi}_i = \frac{1}{v_0} \left( F_{i,\phi}^{(wall)} + F_{i,\phi}^{(int)} + \eta_{i,\phi} \right), \tag{5}$$

where $\mathbf{F}_i^{(wall)}$ stands for the interactions of fish $i$ with the wall, $\mathbf{F}_i^{(int)}$ denotes the total interactions of fish $i$ with all the other fish. The subscript $\phi$ stands for the projection of the respective forces in the turning direction

$$\mathbf{e}_{i,\phi} = -\sin \phi_i \mathbf{e}_x + \cos \phi_i \mathbf{e}_y. \tag{6}$$

The term $\eta_{i,\phi}$ stands for a zero-mean Gaussian noise with $\langle \eta_{i,\phi}(t)\eta_{j,\phi}(t+\tau)\rangle = 2D_\phi \delta_{ij}\delta(\tau)$. We use as our length unit $L$ the average body length of the zebrafish 30 mm and use the average speed of zebrafish $v_f = 100 \text{ mms}^{-1}$ to derive the time unit as $T = L/v_f = 0.3$ s. In these units we have $D_\phi = 0.015$ and $v_0 = 1$.

We assume that the fish are confined in a circular tank of radius $R = 66.7\,L$ and model the wall avoidance interactions $\mathbf{F}_i^{(wall)}$ by a soft repulsive potential. The interactions between the fish $i$ and any other fish $j$ are assumed to consist of repulsion-attraction $\mathbf{f}_{ij}^{(att)}$, alignment $\mathbf{f}_{ij}^{(al)}$ and hydrodynamic interactions $\mathbf{f}_{ij}^{(hyd)}$, so that the total interaction force acting on fish $i$ reads

$$\mathbf{F}_i^{(int)} = \sum_{j \neq i} \left( \mathbf{f}_{ij}^{(att)} + \mathbf{f}_{ij}^{(al)} \right) + \mathbf{f}_{ij}^{(hyd)}. \tag{7}$$

For the particular expressions of the aforementioned forces, as well as further details on the numerical simulations, we refer the reader to the SI.

## Reporting summary

Further information on research design is available in the Nature Portfolio Reporting Summary linked to this article.

## Data availability

Representative fish coordinate data generated in this study have been deposited in the Zenodo database under accession code https://doi.org/10.5281/zenodo.10015473. Further data that support the findings of this study are available from the corresponding author upon request. Representative simulation codes are also available at the same doi. Further codes that support the findings of this study are available from A.Z. upon request. Further data that support the findings of this study are available from the corresponding author upon request.

## Code availability

https://doi.org/10.5281/zenodo.10015473. The code for the 3D reconstruction of fish trajectories are hosted on GitHub and archived on Zenodo[65]. Simulation codes are also available at the same doi. Further codes that support the findings of this study are available from A.Z. upon request.

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

## Acknowledgements

All the authors are grateful to Chrissy Hammond for help with the zebrafish husbandry. C.P.R. gratefully acknowledges the Alexander von Humboldt Foundation for the generous bestowal of a Friedrich Wilhelm Bessel award which enabled the work reported here to be carried out. Bob Evans and Nigel Wilding are thanked for discussions. A.Z. acknowledges funding support by the Vienna Science and Technology Fund (WWTF) [10.47379/VRG20002]. Y.Y. gratefully acknowledges the China Scholarship Council for funding. The work of H.L. was supported by the DFG (German Research Foundation) within LO 418/29-1 (project number 522595197).

## Author contributions

A.Z. conceived and performed the computer simulations and developed the simulation perception model. She also analysed co-ordinate data from experiments and simulations and model. Y.Y. designed and performed the experiments and analyzed experimental data. A.Z. and Y.Y. contributed equally to this work. C.P.R. and H.L. supervised the project and directed the research. All authors wrote the manuscript.

## Competing interests

The authors declare no competing interests.

## Ethics approval

The UIN (university ethical approval identifier) granted to this project has a reference/19/050. Further information can be found from the animal service unit (ASU) of the University of Bristol. (http://www.bristol.ac.uk/animal-research/about).
