## [Peer Review File · Nature Communications]

REVIEWER COMMENTS

Reviewer #1 (Remarks to the Author):

I liked this study and think it will make a substantial impact to understanding the collective behaviour of animals in general (not just zebrafish, or even just fish). While the approach taken is one from the physical sciences, the findings will be of widespread interest to biologists studying collective behaviour. This is because we typically assume that inter-individual interactions must change in larger groups (as there are more individuals to interact with) and result in different collective behaviours. While this effect is likely to saturate at large group sizes (a fish is unlikely to be able to tell if it's in a group of 100 or 1000), showing that collective behaviour in large groups is already seen in groups of only 3 is not what I would predict. It is excellent to see that this is done with 3D tracking, as most studies on fish collective behaviour have approximated the system to 2D (with fish in shallow water) for ease of data collection.

I found the paper to be very well written, being clear and well-explained enough that it should be understandable to those with a background in biology while still satisfying those from the physical sciences.

I thought the abstract clearly lays out the approach and findings. However, I found the first three sentences didn't make the work sound important enough and with widespread appeal to justify publication in a high impact journal with a broad readership. The authors could, for example (there are other ways to do this), first state how collective movement is key to group living in a diverse range of animals, then how much research has been dedicated to deciphering the inter-individual interactions within groups, and then establish the research question of how these interactions scale with group size, stating that this is poorly understood.

At the start of the "Brief methodology", it would be useful to explain why a "a bowl shaped tank with parabolic section" was used, especially given that the authors later demonstrate that the geometry within which the fish can move is important.

Please explain briefly but explicitly what "lab reference frame" and "reference frame of fish i " are; this could be done in the figure 2 legend.

Fig. 3 legend: It wasn't clear to me what the grey contour lines behind the fish trajectories represented.

I think the label in the figure 4 legend, “(d) Probability distribution of the three-fish bond angle”, should be labelled (c)? Also, further explanation of what “three-fish bond angle” is would be worth adding.

It would benefit the paper to include in the Conclusion a discussion of the biological implications of the findings, especially “nearest neighbor interactions dominate and the other fish are less important”. This would only require a paragraph, but could include the importance for predation risk, as this result implies that during predatory attacks that disturb the structure of large groups, the fish do not need to change the neighbours that they are interacting with. For example, for an analysis of fish collective behaviour during attacks from predators, see

Romenskyy, Maksym, et al. "Quantifying the structure and dynamics of fish shoals under predation threat in three dimensions." *Behavioral Ecology* 31.2 (2020): 311-321

Also, it is worth commenting on the sensory basis of collective behaviour, e.g.

Pita, Diana, et al. "Vision in two cyprinid fish: implications for collective behavior." *PeerJ* 3 (2015): e11113, i.e. whether the lack of longer-range interactions is due to a sensory or cognitive constraint of the fish.

In the Supplementary Information, “not an actual phase transitions” needs correcting.

Christos Ioannou

University of Bristol

Reviewer #2 (Remarks to the Author):

This manuscript provides nice methodological expansions of our understanding of collective behavior of fish. However, while there are several nice aspects provided, I have concerns about the methods and experimental design that prevent me to recommend acceptance for publication in its current form.

1. Sample size for each 2, 3, 4 and 50 group size is not provided. Further, the authors state that the same fish were sampled repeatedly (“brief methodology section”). This would mean the data contains either a lot of pseudo-replicates (if group sizes would have been sampled more than once) or no replicate at all (if each group size has been sampled only once) - both is problematic.

2. How are fish identified during tracking and during preceding and subsequent handling? In the methods section, the authors say “Typically, three fish (fish A, B, and C) will be transferred to a temporary tank. The fish A and B will be introduced to the observation tank, where we carried out the

first two-fish observation. Then fish B will be taken back to the temporary tank, while fish C being introduced to the observation tank, so that the second two-fish experiment could be carried out. Finally, all the fish were placed in the observation tank, and we perform the three-fish observation." This is only possible when fish would have been identified and IDs were traced throughout the experiment. Furthermore, this procedure would treat each of the fish in the group differentially thus introducing confounding variation among the three fish. How often was each fish repeatedly tested and how long was a recording session?

How were fish IDs kept during tracking? As far as I understand, the 3D tracking has been done by combining 3 2D-tracks. This would mean that fish IDs in all three 2D tracks must have been traced throughout the recording period - for sometimes 50 individuals! Please provide proof that this has been achieved. If IDs regularly jump or switch, I don't see how the data then can be used to inform any model or interpretation.

3. The simulation model assumes constant speed "We assume that the fish move in 2D with a constant speed v_0 and can only change their velocity orientation given by the angle i (Fig. 2 (a))." It is known that fish use speed adjustments during social interactions and more recent models already include this feature.(Herbert-Read et al. 2011, Herbert-Read et al. 2017, Jolles et al. 2017, Jolles et al. 2020, Klamser et al. 2021). Why isn't variable speed included in the model?

Herbert-Read, J. E., A. Perna, R. P. Mann, T. M. Schaerf, D. J. T. Sumpter, and A. J. W. Ward. 2011. Inferring the rules of interaction of shoaling fish. *Proceedings of the National Academy of Sciences* 108:18726-18731.

Herbert-Read, J. E., E. Rosén, A. Szorkovszky, C. C. Ioannou, B. Rogell, A. Perna, I. W. Ramnarine, A. Kotrschal, N. Kolm, J. Krause, and D. J. T. Sumpter. 2017. How predation shapes the social interaction rules of shoaling fish. *Proceedings of the Royal Society B: Biological Sciences* 284.

Jolles, J. W., N. J. Boogert, V. H. Sridhar, I. D. Couzin, and A. Manica. 2017. Consistent Individual Differences Drive Collective Behavior and Group Functioning of Schooling Fish. *Current Biology* 27:2862-2868.e2867.

Jolles, J. W., N. Weimar, T. Landgraf, P. Romanczuk, J. Krause, and D. Bierbach. 2020. Group-level patterns emerge from individual speed as revealed by an extremely social robotic fish. *bioRxiv:2020.2006.2010.143883*.

Klamser, P. P., L. Gómez-Nava, T. Landgraf, J. W. Jolles, D. Bierbach, and P. Romanczuk. 2021. Impact of Variable Speed on Collective Movement of Animal Groups. *Frontiers in Physics* 9.

4. Please provide also the UIN (university ethical approval identifier).

Reviewer #3 (Remarks to the Author):

The authors investigate the motion of groups of Zebrafish of sizes 2, 3, 4, and 50 and identify three collective patterns: schooling, milling, and swarming. They discuss many-body spatial correlation and group size effects. The study involves experiments and agent-based models. They conclude that Zebrafish interact with their nearest neighbors and that groups of 3 already exhibit the three collective patterns mentioned above.

This is a problem of timely interest, but there are various issues that need to be clarified/addressed.

1. There are several works on the collective motion of animal systems that are comparable to this one. I could not find what distinguishes this study from previous ones. What is the relevant, distinct take-home message here?
2. It is not clear to me how parameters have been selected. How many parameters are there? Is there a quantitative fair comparison between the model and experiments?
3. Are there alternative models/explanations? Small groups of animals were analyzed with simpler models. As in this study, it was found that the motion of individuals is fundamentally given by an attraction mechanism towards the nearest neighbors in the field of view and that leads to a strong velocity correlation among the group members. Interactions are strongly non-reciprocal and there is a front-back asymmetry. See Gomez-Nava et al. *Nature Physics* 18, 1494-1501 (2022). In this work, intermittent behavior refers to the fact the group moves and stops. Can one speculate that a model with fewer parameters can also reproduce Zebrafish data?
4. In previous fish models (see Turnstrom et al. *PLoS Comp. Biol.* 9, e1002925 (2013)) was shown that spontaneous transitions between schooling, milling, and swarming, without varying parameter values, occur. Is this also occurring here?

Response to reviewer one

We thank the reviewer for kindly taking the trouble to read our manuscript and especially for their very positive assessment of our work. We have considered carefully the criticism of the reviewer and we have implemented their helpful suggestions in our revised manuscript. We hope that the reviewer finds our revised manuscript to be suitable for publication in *Nature Communications*.

Reviewer #1 (Remarks to the Author):

I liked this study and think it will make a substantial impact to understanding the collective behaviour of animals in general (not just zebrafish, or even just fish). While the approach taken is one from the physical sciences, the findings will be of widespread interest to biologists studying collective behaviour. This is because we typically assume that inter-individual interactions must change in larger groups (as there are more individuals to interact with) and result in different collective behaviours. While this effect is likely to saturate at large group sizes (a fish is unlikely to be able to tell if it's in a group of 100 or 1000), showing that collective behaviour in large groups is already seen in groups of only 3 is not what I would predict. It is excellent to see that this is done with 3D tracking, as most studies on fish collective behaviour have approximated the system to 2D (with fish in shallow water) for ease of data collection.

I found the paper to be very well written, being clear and well-explained enough that it should be understandable to those with a background in biology while still satisfying those from the physical sciences.

I thought the abstract clearly lays out the approach and findings. However, I found the first three sentences didn't make the work sound important enough and with widespread appeal to justify publication in a high impact journal with a broad readership. The authors could, for example (there are other ways to do this), first state how collective movement is key to group living in a diverse range of animals, then how much research has been dedicated to deciphering the inter-individual interactions within groups, and then establish the research question of how these interactions scale with group size, stating that this is poorly understood.

We thank the reviewer for their kind and highly positive assessment of our manuscript. We are also grateful for the constructive criticism that they offer here and have restructured the introduction accordingly.

At the start of the "Brief methodology", it would be useful to explain why a "a bowl shaped tank with parabolic section" was used, especially given that the authors later demonstrate that the geometry within which the fish can move is important.

This geometry was selected because we want to measure the 3D behaviour of the zebrafish, since the fish naturally swim in 3D in rivers [Shelton, *et al* Zebrafish 17, no. 4 (2020)]. To capture the 3D movement of the fish with a three camera setup, we need all three cameras to view the fish without any obstacles. Therefore, the container should not be a box, in which the corners will be invisible for some cameras. We have made this point clearly in the revised manuscript.

Please explain briefly but explicitly what "lab reference frame" and "reference frame of fish i" are; this could be done in the figure 2 legend.

Thank you for this comment. We have clarified what we mean by laboratory and fish reference frames in the caption to Fig. 2.

Fig. 3 legend: It wasn't clear to me what the grey contour lines behind the fish trajectories represented.

Thank you for this comment — the grey contours have been clarified in the caption.

I think the label in the figure 4 legend, "(d) Probability distribution of the three-fish bond angle", should be labelled (c)? Also, further explanation of what "three-fish bond angle" is would be worth adding.

Thank you for kindly pointing out this typo, it has been fixed.

It would benefit the paper to include in the Conclusion a discussion of the biological implications of the findings, especially "nearest neighbor interactions dominate and the other fish are less important". This would only require a paragraph, but could include the importance for predation risk, as this result implies that during predatory attacks that disturb the structure of large groups, the fish do not need to change the neighbours that they are interacting with. For example, for an analysis of fish collective behaviour during attacks from predators, see Romenskyy, Maksym, et al. "Quantifying the structure and dynamics of fish shoals under predation threat in three dimensions." Behavioral Ecology 31.2 (2020): 311-321 Also, it is worth commenting on the sensory basis of collective behaviour, e.g. Pita, Diana, et al. "Vision in two cyprinid fish: implications for collective behavior." PeerJ 3 (2015): e11113, i.e. whether the lack of longer-range interactions is due to a sensory or cognitive constraint of the fish.

Thank you for this suggestion. We have included a paragraph in the conclusions, which we believe goes towards addressing the biological implications of our findings.

In the Supplementary Information, "not an actual phase transitions" needs correcting.

Thank you — this has been addressed.

Reviewer #2 (Remarks to the Author):

We thank the reviewer for their careful reading of our manuscript and for their thoughtful and constructive criticism. We are also grateful for their positive assessment of the potential for our manuscript. We have endeavoured improve the manuscript in line with the reviewer's comments. In particular, we have extended or model to include variable speed in the agents and include the results in our revised manuscript. We hope that this revised manuscript meets the reviewer's expectations and that is suitable for publication in *Nature Communications*.

This manuscript provides nice methodological expansions of our understanding of collective behavior of fish. However, while there are several nice aspects provided, I have concerns about the methods and experimental design that prevent me to recommend acceptance for publication in its current form.

1. Sample size for each 2, 3, 4 and 50 group size is not provided. Further, the authors state that the same fish were sampled repeatedly ("brief methodology section"). This would mean the data contains either a lot of pseudo-replicates (if group sizes would have been sampled more than once) or no replicate at all (if each group size has been sampled only once) - both is problematic.

Thank you for pointing this out. The sample sizes are as described below, and the following text has been added to the methods section of the revised manuscript.

"The experimental data were sampled as follows. For the two fish experiment, we selected 12 different pairs randomly chosen from a group of 50, and observed each pair for one hour. For the three fish experiment, we selected 6 different triplets from a group of 50, and observed each triplet for one hour. For four fish experiment, we selected 5 different quadruplets from a group of 50, and observed each quadruplet for one hour. The group of 50 itself was observed for one hour."

In this way, our repeated measurements of a given group size sampled different fish, rather than taking multiple measurements of the same fish so that our analysis are not specific to any group. We acknowledge the lack of detailed description on the sample size and we have included the information in the updated manuscript.

2. How are fish identified during tracking and during preceding and subsequent handling? In the methods section, the authors say "Typically, three fish (fish A, B, and C) will be transferred to a temporary tank. The fish A and B will be introduced to the observation tank, where we carried out the first two-fish observation. Then fish B will be taken back to the temporary tank, while fish C being introduced to the observation tank, so that the second two-fish experiment could be carried out. Finally, all the fish were placed in the observation tank, and we perform the three-fish observation." This is only possible when fish would have been identified and IDs were traced throughout the experiment. Furthermore, this procedure would treat each of the fish in the group differentially thus introducing confounding variation among the three fish. How often was each fish repeatedly tested and how long was a recording session?

Thank you for pointing out the issue with the identity of the fish and we certainly agree that more clarification would help. We did not use actual markers to identify the fish, and we always choose the fish with no preference.

However, our fixed recording procedure enables us to label and track the fish in each individual experiment. Operationally, we carry out the following tasks in a typical experiment.

[1] We set up the observation room to have a fixed brightness (~25 LUX) and the water in the tank to have a fixed temperature (~25 C).

[2] We randomly select three fish out of their living tanks, and transfer them to the observation room.

[3] We randomly select two fish out of three for the first observation, and place these two into the observation tank. We label the fish that was not chosen as fish C.

[4] We start a recording timer, and then leave the observation room. The recordings automatically started 10 minutes later. The movements of the fish are recorded for 1 hour at a frame rate of 15fps. Therefore we capture 54000 frames in total in the MP4 video format.

[5] We return to the observation room, and randomly choose one fish in the observation tank, and place it back to a temporary tank. This fish, being taken out from observation, is fish B, and the fish left for observation is fish A. We then place fish C into the observation tank to observe fish A and fish C. And then we repeat step [4].

[6] We return to the observation room and place fish B into the observation tank, and then repeat step [4] to observe the triplet of fish A, B and C.

[7] We return to the observation room, and repeat step [4] again to capture another 3-fish video.

[8] We return to the observation room to place the fish back to their living tank, and use automatic script to process the videos.

We repeated our observations 6 times, and concatenated all the results. The dates of the experiments are as follows: 2021-05-11, 2021-05-12, 2021-05-14, 2021-05-18, 2021-05-19, 2021-06-01.

Since we always perform selection without preference, we do not know exactly how many times one fish was selected. There are 50 fish in total, therefore the probability of one fish being selected twice is 1.4%. We acknowledge that our previous manuscript lack the detail of the experimental procedure. This description has been added to the SI of our revised manuscript.

How were fish IDs kept during tracking? As far as I understand, the 3D tracking has been done by combining 3 2D-tracks. This would mean that fish IDs in all three 2D tracks must have been traced throughout the recording period - for sometimes 50 individuals! Please provide proof that this has been achieved. If IDs regularly jump or switch, I don't see how the data then can be used to inform any model or interpretation.

This is a very good point, and we acknowledge that we should have been rather more clear in the manuscript. We do not have a 100% confidence in the fish IDs during the tracking [our confidence is around 90% for a group of 50 fish and higher for smaller groups, see below and Yang *et al. PLOS Computational Biology* 18 14 (2022)]. In some circumstances, this can be a significant issue which limits determination of correlation functions which directly require the IDs being consistently correct, for instance the time correlation function [Nagy *et al Nature* 464 890 (2010)]. However, for the results presented in our current manuscript, we only utilized the positions and velocities of the fish, so the switching of IDs is not a major issue here.

In fact we did not get 3D trajectories by combining 2D trajectories. Instead, we first obtain 3D coordinates of the fish, and then link these locations into trajectories. This association process (to find the same fish in pictures taken from different cameras) can be carried out just by considering the multi-view geometry, as well as the refraction of water. We then followed the existing Lagrangian particle tracking algorithm to link these positions into trajectories [Ouellette *et al. Experiments in Fluids* 40 301 (2005); Xu *Measurement Science and Technology* 19 075105 (2008)].

The linking process is not perfect as we encounter breaking of trajectories, exactly because IDs of the fish are not followed correctly all the time. Even though we are aware of existing software to better handle IDs of individuals, it is hard for us to use such "conventional" 2D tracking software because we used a multiple-camera setup to reconstruct 3D trajectories. In our case, the relative distances between the fish and the cameras change frequently so the size and shape of the fish are not consistent. The changing shapes break the assumption of IdTracker [Pérez-Escudero *et al. Nature Methods* 11 743 (2014)] and similar software to the best of our knowledge.

To tackle the 3D tracking task, we developed our own tracking system all from scratch, and we briefly mentioned the system in our previous publication [Yang *et al. PLOS Computational Biology* 18 14 (2022)]. We also shared all the related source code on GitHub (<https://github.com/yangyushi/FishPy>).

We are confident that our camera system and tracking code produce good results, because we tested them with simulated data. As described in the SI of our previous paper [Yang *et al. 2022*], we simulated the trajectories of 50 Vicsek agents and rendered these agents in simulated conditions mimicking our experimental setup. Knowing the ground truth of the movement of these agents, we re-constructed the movements with our algorithms. By comparing the ground truth with our reconstruction, we learned that the algorithm locates 45 fish out of fifty. The missing fish would cause IDs to change and trajectories to break, but have a limited impact on the calculated order parameters and correlation functions.

Furthermore, we limited our analysis to functions that only require the locations and velocities of the fish group, regardless of the fish ID, as shown in equations (1) and (2) in the main text. Therefore, the problem of fish being assigned an incorrect ID only enters the calculation indirectly by causing the velocities to be inaccurate. For example, If fish A in time t was assigned to fish B in time $t+1$, then the only error affecting the analysis is a wrong velocity vector in time t . Since we repeated carried out long observations, we have large enough data to ensure these infrequent error not contributing significantly.

We have addressed this point in the revised manuscript and added a discussion to the SI.

*3. The simulation model assumes constant speed “We assume that the fish move in 2D with a constant speed v_0 and can only change their velocity orientation given by the angle in (Fig. 2 (a)).” It is known that fish use speed adjustments during social interactions and more recent models already include this feature.(Herbert-Read *et al. 2011*, Herbert-Read *et al. 2017*, Jolles *et al. 2017*, Jolles *et al. 2020*, Klamser *et al. 2021*). Why isn't variable speed included in the model?*

Thank you for the comment. It is quite true that including variable speed in the model has the potential to influence our results. And, as the reviewer points out, there are examples in the literature where differences between individuals in biological systems have been studied. We have therefore incorporated these effects into our model, by allowing variable speeds among the agents. We have now performed new simulations with variable speed and analysed these. Under the observables that we use, in fact this does not massively change the outcome. We discuss this new aspect of the study in our revised manuscript in the context of the references the the referee has kindly provided and other work in the literature.

4. Please provide also the UIN (university ethical approval identifier).

The UIN granted to our project has a reference UB/19/050. Further information can be found from the animal service unit (ASU) of the University of Bristol. (<http://www.bristol.ac.uk/animal-research/about>). This is now in the revised manuscript.

Reviewer #3 (Remarks to the Author):

We thank the reviewer for their careful reading of our manuscript and for their thoughtful and constructive criticism. We are also grateful for their positive assessment of the potential for our manuscript. We have endeavoured improve the manuscript in line with the reviewer's comments and we hope that the revised manuscript meets their expectations and they find it suitable for publication in *Nature Communications*.

The authors investigate the motion of groups of Zebrafish of sizes 2, 3, 4, and 50 and identify three collective patterns: schooling, milling, and swarming. They discuss many-body spatial correlation and group size effects. The study involves experiments and agent-based models. They conclude that Zebrafish interact with their nearest neighbors and that groups of 3 already exhibit the three collective patterns mentioned above.

This is a problem of timely interest, but there are various issues that need to be clarified/addressed.

1. There are several works on the collective motion of animal systems that are comparable to this one. I could not find what distinguishes this study from previous ones. What is the relevant, distinct take-home message here?

Thank you for this question. The central take-home message, which we believe distinguishes our work from others is that we explicitly consider higher-order spatial correlations and the effect of group size, both as isolated groups and as sub-groups of a larger group. While it is not clear that simple approximations to higher-order spatial correlations would work, namely that that 2-body correlations can be used to predict 3-body correlations, to a reasonable extent this turns out to hold. Our emphasis on group size shows that there is no "jump" in the observables we consider upon changing the group size, instead these change gradually. Most surprisingly, we find that under our analysis, small groups of fish within a rather larger are almost indistinguishable from isolated small groups of the same size.

We have made this more clear in the revised manuscript. In particular, we have very substantially rewritten the introduction, to explain how our study is distinct and to emphasize the scientific questions that it addresses.

We have further added additional references [for example Ranganathan, et al *ArXiv* 2211.04352 (2022) and Gomez-Nava, et al *Nature Phys.* 18, 1494–1501 (2022)], and references suggested by reviewer #2, such as Klamser et al. *Front. Phys.* 9, 715996 (2021) and explained the distinction of our work from these other contributions.

2. It is not clear to me how parameters have been selected. How many parameters are there? Is there a quantitative fair comparison between the model and experiments?

Thank you for this point. In fact our approach is more qualitative. We are looking for a model with the same generic behaviour, rather than a precise parameterisation to the experimental zebrafish system. We mention that a more quantitative approach would be an intriguing further study in the in our discussion of the simulation results.

We have justified the selection of our parameters in the SI and listed the values that we use in our simulations in supplementary tables 1 and 2 for an easy reference

3. Are there alternative models/explanations? Small groups of animals were analyzed with simpler models. As in this study, it was found that the motion of individuals is fundamentally given by an attraction mechanism towards the nearest neighbors in the field of view and that leads to a strong velocity correlation among the group members. Interactions are strongly non-reciprocal and there is a front-back asymmetry. See Gomez-Nava et al. *Nature Physics* 18, 1494-1501 (2022). In this work, intermittent behavior refers to the fact the group moves and stops. Can one speculate that a model with fewer parameters can also reproduce Zebrafish data?

Thank you for enquiring as to why the model needs this many parameters. We should have been more clear in the previous manuscript. In fact, in a previous manuscript [Yang et al. *PLOS Computational Biology* 18 14 (2022)], some of us explored a simpler model, however by construction the modified Vicsek used there does not capture spatial correlations accurately. Other possibilities include a zonal model [Couzin et al. *J. Theo. Biol.* 218, 1 (2002)], but this too by construction does not capture the spatial correlations that we study here. We discuss this point when we consider the computer simulations in detail in the SI.

As we argue in the SI, while attraction/repulsion in the field of view is the dominant interaction in our system, it fails to capture the milling state, found in the experiment for group sizes $N \geq 3$. In order to enhance its occurrence usually the strength — or range — of alignment interactions is increased, as e.g. in the case of the zonal model. This, however, leads to spatial correlations very different from those encountered in the experiments. We have found that this issue can be circumvented by accounting for some simplified form of hydrodynamic interactions, described in the SI. From our point of view therefore, our model with parameters for the strength and range for the basic interactions (attraction, alignment, hydrodynamic) and wall avoidance is a relatively simple model that can capture satisfactorily both the intermittent dynamics and the predominant features of the spatial correlations found in our experiment.

4. In previous fish models (see Turnstrom et al. *PLoS Comp. Biol.* 9, e1002925 (2013)) was shown that spontaneous transitions between schooling, milling, and swarming, without varying parameter values, occur. Is this also occurring here?

Thank you for this comment. Indeed, one set of parameters were used (specified in Supplementary Tables 1 and 2 in the revised manuscript) with which the simulated system exhibited spontaneous transitions between the three states, milling, swarming and schooling similar to the work by Turnstrom et al [*PLoS. Comp. Biol.* 9 e1002915 (2013)]. We make this point clearly in the revised manuscript.

REVIEWERS' COMMENTS

Reviewer #1 (Remarks to the Author):

The authors have addressed my comments adequately and I would recommend publication.

Reviewer #3 (Remarks to the Author):

The authors have addressed all my comments and questions, and modify accordingly not only the main text, but also the SI. I think the contribution of the authors in the revised version has become clear. It is always difficult to determine whether a work deserves to be published in a journal as Nature Comm. But it is evident that the authors have done an effort to answer my questions/critiques. It remains, however, a point that I find suspicious, and the authors could easily improve: there is only one movie in the SI.

Reviewer #3 (Remarks to the Author):

The authors have addressed all my comments and questions, and modify accordingly not only the main text, but also the SI. I think the contribution of the authors in the revised version has become clear. It is always difficult to determine whether a work deserves to be published in a journal as Nature Comm. But it is evident that the authors have done an effort to answer my questions/ critiques. It remains, however, a point that I find suspicious, and the authors could easily improve: there is only one movie in the SI.

We thank the review for kindly taking the trouble to assess our manuscript. Further to the reviewer's comments, we have added a further three movies which we believe exemplify a key point of our manuscript namely the behaviour of the zebrafish system as the system size is changed.